# High-Efficiency Photo-Fenton-like Catalyst of FeOOH/g-C_3_N_4_ for the Degradation of PNP: Characterization, Catalytic Performance and Mechanism Exploration

**DOI:** 10.3390/molecules29133202

**Published:** 2024-07-05

**Authors:** Rongjun Su, Junhao Wang, Hao Jiang, Lan Wei, Deying Mu, Chunyan Yang

**Affiliations:** 1School of Food Engineering, Harbin University of Commerce, Harbin 150076, China; wjh3097451424@163.com (J.W.); wjh343856092@163.com (H.J.); 18845111454@163.com (L.W.); mudeying2004@163.com (D.M.); 2College of Architecture and Environment, Institute of New Energy and Low-Carbon Technology, Sichuan University, Chengdu 610207, China

**Keywords:** FeOOH/g-C_3_N_4_, photo-Fenton-like, visible light, PNP

## Abstract

The composite photocatalyst FeOOH/g-C_3_N_4_ was prepared through thermal polycondensation and co-precipitation methods, followed by XRD, SEM and UV-vis characterization. The stability of FeOOH/g-C_3_N_4_ was explored by the recycling test. The active species in the reaction system were investigated by the capture experiment. The results indicated that the optimal preparation condition for g-C_3_N_4_ involved calcination at 600 °C for 4 h. XRD analysis revealed that g-C_3_N_4_ exhibits a high-purity phase, and Fe in FeOOH/g-C_3_N_4_ exists in a highly dispersed amorphous state. SEM analysis showed that FeOOH/g-C_3_N_4_ has a rough surface with an irregular layered structure. Element composition analysis confirmed that the content of elements in the prepared catalyst is consistent with the theoretical calculation. FeOOH/g-C_3_N_4_ possesses the largest specific surface area of 143.2 m^2^/g and a suitable pore distribution. UV-vis DRS analysis showed that the absorption intensity of FeOOH/g-C_3_N_4_ is stronger than that of g-C_3_N_4_. When the catalyst dosage was 1.0 g/L, the H_2_O_2_ dosage was 4 mmol/L, the PNP initial concentration was 10 mg/L and the initial pH value was 5, the PNP removal could reach 92% in 120 min. Even after 5 cycles, the efficiency of PNP removal by FeOOH/g-C_3_N_4_ remains nearly 80%. The capture experiment indicated that both •OH and •O_2_^−^ play roles in the photocatalytic degradation of PNP, with •OH being more significant. These findings affirm that FeOOH has been successfully incorporated into g-C_3_N_4_, resulting in a conspicuous catalytic effect on the degradation of PNP in the visible light-assisted Fenton-like reaction.

## 1. Introduction

Advanced oxidation processes (AOPs) have valuable practical applications because of their strong mineralization ability, fast reaction speed, easy operation and simple process and equipment [1]. What is more, Fenton-like advanced oxidation processes have been widely used due to their advantages of high natural abundance, cost-effectiveness, low toxicity and environmental friendliness [2,3,4,5]. This process has been proven to be efficient in the degradation of organics with iron-based multiphase catalysts such as Fe_3_O_4_ and *α*-FeOOH in wastewater [6,7]. Among them, hydroxyl iron oxide (FeOOH) has the advantages of stable chemical properties, high specific surface area, wide pH range and controllable iron solution, which has been reported in many studies [8,9,10,11,12].

Traditional semiconductor photocatalysts, such as TiO_2_- [13,14] and ZnO-based [15,16] catalytic materials, have high photocatalytic activity. However, they have a low utilization ratio of sunlight in the reaction process because they had wide band gaps and can only respond to UV energy, which is less than 5% of the total energy of sunlight. As a metal-free semiconductor with a layered structure, g-C_3_N_4_ has a band gap of about 2.7 eV and a strong visible light-absorption capacity [17]. Since its discovery in 2009, g-C_3_N_4_ has gradually become a research hotspot [18]. However, g-C_3_N_4_ has a narrow band gap and photogenerated electron–hole pairs are easy to recombine, which limits its application in the field of photocatalysis [19,20]. Introducing element doping [21,22] and heterojunction construction [23,24] can make the catalyst structure form defects, which can act as a trap for photogenerated electrons and promote the separation of photoexcited carriers [25]. Simultaneously, impurity energy levels can be generated in the energy gap to change the band gap structure of semiconductor materials, so as to improve the redox ability of the valence band (VB) and conduction band (CB), making the band gap structure more suitable for photocatalytic reaction [26,27]. FeOOH, with the advantages of a narrow band gap and a broad visible absorption region, has been used for coupling with other semiconductors for applications such as direct photocatalytic degradation of organic matter [28]. In addition, the band gap of amorphous FeOOH is much smaller than that of the corresponding crystalline materials, which results in a broader visible absorption region and excellent photocatalytic performance [29]. However, there are few reports on amorphous FeOOH as a photocatalyst modifier.

P-nitrophenol (PNP) usually exists in sewage, natural water bodies and soil. It can enter the human body through oral, respiratory and skin contact. Additionally, PNP has the potential to accumulate in animal and human bodies through the food chain. PNP structure is relatively stable and widely used as a raw chemical material, such as medicine, dyes and pesticides [30]. Due to the presence of nitro in its molecular structure, it is difficult to biodegrade in nature, so it is extremely stable in the environment [31]. Prolonged exposure to drinking water contaminated with PNP may lead to adverse health effects, including dizziness, memory loss and neurological diseases [32]. Therefore, it is very important to remove PNP from water by efficient and cost-effective means. Many researchers have focused on the development of novel catalysts with high degradation efficiency for PNP.

Based on the above research basis, FeOOH/g-C_3_N_4_ composite visible light catalyst was prepared by thermal polycondensation and co-precipitation method in this project. With PNP in water as the main target pollutant, the photo-assisted Fenton oxidation system was established to study the visible light catalytic performance, and the mechanism of FeOOH/g-C_3_N_4_ visible light catalytic on the degradation of PNP was discussed.

## 2. Experimental

### 2.1. Materials

All reagents used in the experiment are analytically pure, including urea, ferric chloride hexahydrate, ammonium bicarbonate, anhydrous ethanol, sodium hydroxide, hydrochloric acid (Xilong Scientific Co., Ltd., Shantou, China), 30% hydrogen peroxide (Tianjin Continental Chemical Reagent Factory, Tianjin, China), etc.

### 2.2. Preparation of FeOOH/g-C_3_N_4_

Preparation of g-C_3_N_4_: 10 g of urea was put into a crucible, which was then fully covered and wrapped with aluminum foil. Subsequently, the crucible was placed in a muffle furnace. The initial temperature was set at 20 °C. The crucible was heated to different preset temperatures and was maintained at each temperature for 4 h, and then cooled to room temperature.

The preparation of FeOOH/g-C_3_N_4_: 0.3 g of g-C_3_N_4_ was taken into 40 mL anhydrous ethanol, and subjected to ultrasonic treatment (Type KQ5200B ultrasonic cleaner, Kunshan ultrasonic instrument Co., Ltd., Shanghai, China) for 1 h. Subsequently, 0.39 g of FeCl_3_·6H_2_O was added with ultrasonic treatment for 10 min. Following this, 0.343 g of NH_4_HCO_3_ was added, and then the mixture underwent magnetic stirring for 8 h. The resulting mixture was subjected to centrifugal washing three times and then dried at 50 °C. The resulting catalyst is denoted as x-FeOOH/g-C_3_N_4_ (x represents the mass fraction of FeOOH). As a control, pure FeOOH without g-C_3_N_4_ was synthesized in the same procedure.

### 2.3. Characterization

Phase analysis of the catalyst was carried out by X-ray diffraction (XRD, D/Max-RB). The micromorphology of the catalyst was analyzed by scanning electron microscope (SEM, Sigma 500, Sigma America, Ronkonkoma, NY, USA). The X-ray energy dispersive spectrometer (EDS) was used to analyze the composition of elements in the micro-structure of catalysts. The Brunauer–Emmett–Teller method (BET, ASAP2460, Micromeritics, Norcross, GA, USA) was used to analyze the specific surface area and pore distribution of the catalysts. Ultraviolet-visible diffuse reflection spectroscopy (UV–Vis DRS, Lambda 950, PerkinElmer, Waltham, MA, USA) was applied to probe the optical performance of the catalysts.

### 2.4. Experimental Procedures

The reaction device used in the photocatalytic degradation experiment is shown in Figure 1. The optimization of single photocatalyst preparation conditions was achieved by assessing RhB removal. A 0.25 g amount of photocatalysts and 250 mL of RhB solution (10 mg/L) were placed in a photocatalytic reactor and stirred for 30 min under dark conditions to reach an equilibrium of adsorption and desorption. Then, a 500 W xenon lamp (type GXH500 long arc xenon lamp, Shanghai Jiguang Special Lighting electric appliance factory, Shanghai, China) was turned on to radiate the reaction system. Samples of the reaction solution (2–3 mL) were taken out every 30 min, and the clear solution was filtered out through a 0.22 µm membrane. The optimization of composite catalyst-preparation conditions was conducted using PNP as the target pollutant, and the effects of different degradation processes were analyzed using the same method as described above. In addition, 3 mmol/L H_2_O_2_ was added immediately after the light source was turned on to construct the photo-Fenton system, and the concentration of the PNP in the reaction solution was determined at 400 nm at different time intervals.

After the photocatalytic degradation experiment was completed, the remaining reaction liquid was centrifuged in a centrifugal tube to separate solids, washed with deionized water and anhydrous ethanol, and dried for later use. The recovered catalyst was reused in the photocatalytic degradation experiment.

### 2.5. Analytical Methods

UV-vis (Type 722N visible spectrophotometer, Shanghai Yuanxi Instrument Co., Ltd., Shanghai, China) was used to determine the concentration of target pollutants in the reaction solution, and the RhB maximum absorption wavelength of 554 nm.

Isopropanol (IPA), ethylenediamine tetraacetic acid disodium (EDTA-2Na) and *p*-benzoquinone (BQ) were selected as the capture agents of •OH, h^+^ and•O_2_^−^, respectively. The experimental procedure was the same as in Section 2.4. After the system reached the adsorption and desorption equilibrium, a certain amount of the aforementioned trapping agent is added into the system to continue the photocatalytic degradation experiment.

## 3. Results and Discussion

### 3.1. Evaluation of FeOOH/g-C_3_N_4_ Catalytic Performance in Different Preparation Conditions

Figure 2a shows the effect of calcination temperature on the photocatalytic degradation of RhB by g-C_3_N_4_. The results show that the degradation rate of RhB was about 95% when the calcination temperature was 600 °C. This result is because the C/N ratio of g-C_3_N_4_ increases with the increase in temperature, and the band gap decreases according to other references [33,34]. Generally, the narrower the band gap, the larger the corresponding light absorption range. But an excessively narrow band gap may facilitate the recombination of photogenerated electron–hole pairs, resulting in low light quantum utilization and a weakened redox capacity. The catalyst prepared at 600 °C may combine these two effects to achieve the most suitable catalytic performance. Therefore, 600 °C was chosen as the optimal calcination temperature.

As shown in Figure 2b, the degradation efficiency of RhB was about 95% when the calcination time was 4 h. This phenomenon can be attributed to the increase in the specific surface area and pore volume of g-C_3_N_4_ with prolonged calcination time [35]. However, once the calcination time surpasses a certain threshold, the structure tends to stabilize, leading to no significant alteration in the photocatalytic capability of g-C_3_N_4_. Therefore, 4 h was chosen as the optimal calcination time.

The FeOOH/g-C_3_N_4_ photocatalyst prepared with different mass fractions of FeOOH was used in photocatalytic degradation of PNP for 120 min. The total removal was calculated and the dark adsorption curve was plotted with the results illustrated in Figure 3a,b. Figure 3a shows that the prepared 0.3FeOOH/g-C_3_N_4_ composite photocatalyst has the highest efficiency in PNP degradation, achieving a degradation rate of approximately 82% under visible light for 120 min. This may be because a mass fraction of 0.3FeOOH is the ideal doping amount, resulting in the optimal specific surface area and pore size, thereby enhancing photocatalytic efficacy. Figure 3b shows that all three photocatalysts can reach adsorption equilibrium after a dark reaction for 30 min. Notably, the 0.3FeOOH/g-C_3_N_4_ photocatalyst exhibits the highest adsorption capacity for PNP, with an adsorption ratio of 7.6% after 30 min. This heightened adsorption is likely due to its high specific surface area. In the photocatalytic system, catalysts with large specific surface areas can adsorb more target pollutants, which is conducive to the subsequent photocatalytic reaction. Thus, FeOOH doping enhances the PNP degradation rate of g-C_3_N_4_, and 0.3FeOOH/g-C_3_N_4_ demonstrates superior photocatalytic performance. Therefore, the 0.3FeOOH/g-C_3_N_4_ composite photocatalyst was used in subsequent photocatalytic performance tests and the influence of experimental parameters was assessed.

In conclusion, the optimal preparation conditions of g-C_3_N_4_ were calcination at 600 °C for 4 h. Similarly, for the FeOOH/g-C_3_N_4_ composite photocatalyst, the optimal conditions were determined by adding FeOOH with a mass fraction of 0.3.

### 3.2. Characterizations of FeOOH/g-C_3_N_4_

#### 3.2.1. Crystal Structure Analysis

XRD was used to characterize the crystal structure of the catalysis. As shown in Figure 4, there are two distinct characteristic peaks located at 12.8° and 27.2°, which belong to the (100) and (002) surfaces of g-C_3_N_4_, respectively, indicating that g-C_3_N_4_ has a high purity phase (JCPDS No. 87-1526). There is no obvious FeOOH diffraction peak in the figure for FeOOH/g-C_3_N_4_ and FeOOH, which may be due to the synthesized FeOOH being amorphous.

EDS analysis was performed on the sample of the 0.3FeOOH/g-C_3_N_4_ photocatalyst, and the results are shown in Figure 5. The results show that the prepared catalyst contains 41.9% N, 30.9% C, 15.2% O and 7.0% Fe, which is consistent with the theoretical calculation results.

#### 3.2.2. Microtopography Analysis

Figure 6 displays the SEM microscopic morphology analysis results. The sample of g-C_3_N_4_ presents an irregular layered structure, which corresponds to the traditional two-dimensional layered structure of g-C_3_N_4_. The nanosheets in g-C_3_N_4_ have a smooth surface, while the FeOOH/g-C_3_N_4_ composite photocatalyst has a rough surface. This can be attributed to the generation of granular FeOOH during the co-precipitation process, which uniformly adheres to the surface of g-C_3_N_4_. This phenomenon contributes to an increase in the specific surface area, enhancing the number of active sites on the catalyst surface.

#### 3.2.3. Analysis of Specific Surface Area and Aperture Structure

Brunauer–Emmett–Teller (BET) and Barrett–Joyner–Halenda (BJH) methods were used to analyze the specific surface area and pore distribution of the prepared photocatalysts, respectively, with the results shown in Figure 7. As can be seen from Figure 7a, N_2_ adsorption–desorption isotherms of g-C_3_N_4_ and FeOOH/g-C_3_N_4_ samples are consistent with the hysteresis characteristic curves of mesoporous materials with a type IV isotherm and a type H3 adsorption hysteresis loop in the high *P*/*P*_0_ range. This aligns with the pore size distribution curve in Figure 7b, indicating a mesoporous structure in the catalysts. This indicates that the introduction of FeOOH has no significant effect on pore structure. BET model analysis showed that the specific surface areas of FeOOH, g-C_3_N_4_ and FeOOH/g-C_3_N_4_ were 62.5, 96.9 and 143.2 m^2^/g, respectively. The composite catalyst demonstrated a substantial increase in specific surface area compared to individual FeOOH and g-C_3_N_4_. According to the BJH model, the average pore sizes of FeOOH, g-C_3_N_4_ and FeOOH/g-C_3_N_4_ were 3.406, 3.053 and 3.823 nm, respectively, with corresponding pore volumes of 0.086, 0.214 and 0.849 cm^3^/g. Higher specific surface area enhances adsorption capacity, while a rich pore structure exposes more active sites, thereby improving catalytic activity. In summary, FeOOH/g-C_3_N_4_ with its largest specific surface area and suitable pore distribution combines the advantages of the two single catalysts, exhibiting significant photocatalytic potential. 

#### 3.2.4. Optical Property Analysis

The optical properties of the prepared FeOOH/g-C_3_N_4_ photocatalyst were analyzed by UV-Vis DRS. As shown in Figure 8a, both g-C_3_N_4_ and FeOOH/g-C_3_N_4_ can absorb visible light. The absorption edge of pure g-C_3_N_4_ is about 430 nm, while that of FeOOH/g-C_3_N_4_ is extended to about 480 nm, which means it red-shifted and has a wide range; thus, the absorption intensity is strengthened. This indicates that the prepared FeOOH/g-C_3_N_4_ photocatalyst demonstrates efficient utilization of visible light. For semiconductor materials, the band gap (Eg) is calculated as αhν = A(hν − Eg)^n/2^, where α is the absorption coefficient, h is Planck’s constant, ν is the optical frequency, and A is a constant. n value is determined by the type of semiconductor leap. Since g-C_3_N_4_ is an indirect semiconductor, n is taken as 4, while FeOOH is a direct semiconductor, so n is 1. Figure 8b,c display band gaps of 2.53 eV and 1.90 eV for g-C_3_N_4_ and FeOOH, respectively. The charge transportation between the FeOOH/g-C_3_N_4_ heterojunction follows a Z-scheme according to references [36,37]. Hence, electron–hole (e^−^-h^+^) pairs were generated on the surface of FeOOH and g-C_3_N_4_ photocatalysts under visible light irradiation. Briefly, the electrons are excited into the CB while the holes remain in the VB. Due to the formation of a heterojunction, the FeOOH holes migrate downward. Conversely, the electrons of g-C_3_N_4_ will migrate to the more positively charged CB of FeOOH. As a result, the electrons gather on the CB of FeOOH and the holes gather on the VB of g-C_3_N_4_ for the purpose of photogenerated charge separation.

### 3.3. Catalytic Performance of Different Degradation Processes

The degradation effects of PNP under different degradation processes were analyzed and measured, as shown in Figure 9. As can be seen from the figure, the removal of PNP by light alone is only about 10%, because the structure of PNP is stable under visible light and is difficult to degrade through this process. The addition of H_2_O_2_ to this system enhances the removal of PNP to approximately 43%, indicating that H_2_O_2_ has a certain oxidation effect, but cannot reach the ideal efficiency, because the amount of •OH generated by H_2_O_2_ decomposition under light is still small, thus the oxidation effect on PNP is limited. Based on light, H_2_O_2_ and FeOOH/g-C_3_N_4_, the removal of PNP sharply rises to about 82%, indicating that most organic matter can be rapidly oxidized and decomposed. This indicates that FeOOH/g-C_3_N_4_ has an obvious effect on PNP degradation in the photo-Fenton-like reaction system. The catalytic performance of FeOOH/g-C_3_N_4_ under visible light, combined with the strong oxidation of the Fenton system, achieves an ideal synergistic effect in PNP degradation.

### 3.4. Effect of Reaction Conditions on the Decomposition of PNP

#### 3.4.1. Effect of H_2_O_2_ Dosage

A 1.0 g/L FeOOH/g-C_3_N_4_ photocatalyst was placed into a 250 mL PNP solution with a concentration of 10 mg/L. The system underwent dark adsorption for 30 min until the adsorption equilibrium was reached, and then different concentrations of H_2_O_2_ (2 mmol/L, 3 mmol/L, 4 mmol/L, 5 mmol/L) were added. The influence of catalyst dosage on PNP degradation was studied under simulated visible light radiation for 120 min. The degradation curve is shown in Figure 10a. The pseudo-first-order reaction kinetics was fitted, with the results shown in Figure 10b and Table 1.

As shown in Figure 10a, the degradation rate of PNP in the reaction increases with the increase in H_2_O_2_ dosage. However, when the concentration increases above 4 mmol/L, the PNP removal exhibits marginal improvement, which is mainly because excessive H_2_O_2_ will capture •OH instead and increase side reactions in the system. As a result, the utilization rate of •OH in the system is reduced, which affects the effect of PNP removal. Thus, the optimal concentration is identified as 4 mmol/L. Figure 10b and Table 1 show that the photocatalytic reaction rate enhanced with the increase in H_2_O_2_ dosage, and the pseudo-first-order reaction rate constant is the maximum and remains nearly the same at 4 mmol/L and 5 mmol/L. Therefore, the optimal dosage of 4 mmol/L is selected for further study on the influence of photocatalytic parameters.

#### 3.4.2. Effect of Catalyst Dosage

The influence of catalyst dosage on PNP degradation was studied with various dosages of FeOOH/g-C_3_N_4_ catalyst (0.6 g/L, 0.8 g/L, 1.0 g/L and 1.2 g/L), and other conditions were the same as in Section 3.4.1. The degradation curves are shown in Figure 11a. The quasi-first-order reaction kinetics was fitted, and the results are shown in Figure 11b and Table 2.

As shown in Figure 11a, when the dosage of the catalyst increased from 0.6 g/L to 1.0 g/L, the degradation rate of PNP also increased. This is mainly attributed to the enlarged number of active sites on the catalyst surface and the increased concentration of Fe^3+^. For the constant amount of pollutants, the enhanced catalyst dosage leads to more significant participation of PNP in catalytic oxidation reactions, consequently improving its degradation rate. However, when the dosage of the catalyst continues to increase to 1.2 g/L, the removal rate decreases. The reason is that the excessive catalyst amount could lead to a decrease in light transmittance in the reaction system, thus resulting in insufficient light intensity and inhibitory effects on the photocatalytic effect. When the dosage is 1.0 g/L, the degradation rate of PNP is up to 91% after 120 min. Figure 11b and Table 2 show that a higher dosage of catalyst results in an increased photocatalytic reaction rate with the maximum quasi-first-order reaction rate constant at 1.0 g/L. Therefore, the optimal FeOOH/g-C_3_N_4_ dosage of 1.0 g/L is selected to proceed with the subsequent investigation of the influence of photocatalytic parameters.

#### 3.4.3. Effect of Initial pH Value

NaOH and HCl solution were used to adjust the initial pH value of the reaction system to 3, 5, 7, 9, and other conditions were the same as in Section 3.4.1. The influence of the initial pH value on the degradation of PNP was studied. Figure 12a illustrates the degradation curves. The quasi-first-order reaction kinetics was fitted, and the results are shown in Figure 12b and Table 3.

In Figure 12a, the initial pH value greatly affects the PNP removal, showing a gradual decrease with rising pH value. This indicates that the system has better catalytic activity in an acidic environment, which is the same as the conclusion of other studies in the general heterogeneous Fenton system. Under acidic conditions, Fe^3+^ can participate in the reaction in the form of ions without facile hydrolysis, increasing the utilization rate of •OH in the system, thus improving the PNP removal. In addition, under acidic conditions, H^+^ can promote the decomposition of H_2_O_2_ to produce •OH, which improves the ability of catalytic oxidation of organic pollutants. Conversely, under alkaline conditions, Fe^3+^ is prone to hydrolysis, resulting in a decrease in catalytic activity, and H_2_O_2_ easily loses its activity in alkaline conditions, which reduces the generation of •OH and inhibits the effective degradation of PNP. In addition, it can be seen from the figure that when the pH value is 3 and 5, the final PNP removal is relatively close, which can reach more than 92%. Given the cost-effectiveness and proximity to typical water pH, the optimal pH value of 5 is selected for this experiment. As shown in Figure 12b and Table 3, different initial pH values in reaction systems exhibit a consistent trend in influencing the photocatalytic reaction rate.

#### 3.4.4. Effect of PNP Initial Concentration

The concentration of PNP solution was different (5 mg/L, 10 mg/L, 15 mg/L and 20 mg/L), and other conditions were the same as in Section 3.4.1. The effect of the initial concentration of target substance PNP on its degradation was studied. The degradation curve is illustrated in Figure 13a. The quasi-first-order reaction kinetics was fitted, and the results are shown in Figure 13b and Table 4. As shown in Figure 13a, the total degradation rate decreases with the increase in the initial concentration of PNP. When the concentration of PNP is 5 mg/L, the removal is close to 98% at 120 min, but when the concentration increases to 20 mg/L, the removal rate is only about 63% at 120 min. This is because in high concentrations of PNP, more intermediate products are produced in the photocatalytic reaction process, and these substances will also react with •OH. At the same time, when the dosage of the catalyst is fixed, the number of active sites provided and the number of active groups that can participate in the photocatalytic reaction are limited, which also limits the number of PNP molecules that can participate in the photocatalytic reaction. Therefore, when the concentration of PNP increases, the limited catalyst cannot provide enough reaction sites, and the overall degradation rate will decrease correspondingly. Figure 13b and Table 4 further demonstrate that the photocatalytic reaction rate also decreases with the gradual increase in the initial concentration of PNP.

#### 3.4.5. Stability of FeOOH/g-C_3_N_4_

The FeOOH/g-C_3_N_4_ involved in the photocatalytic reaction was centrifuged, washed, recovered and dried, and its visible light catalytic performance for PNP was tested again to evaluate its recycling performance. As shown in Figure 14, the degradation rate of PNP by FeOOH/g-C_3_N_4_ was nearly 80% even when it was reused five times, which indicates that the FeOOH/g-C_3_N_4_ photocatalyst prepared in this paper has better recycling performance and good stability.

### 3.5. Identification of Active Species in the Photo-Fenton-like Process

In the process of photocatalytic degradation, three active substances, namely •OH, h^+^ and •O_2_^−^, typically play crucial roles. In this paper, isopropyl alcohol (IPA), EDTA-2Na and *p*-benzoquinone (BQ) were selected as the capture agents of •OH, h^+^ and •O_2_^−^, respectively. The effect of PNP degradation in each capture experiment was compared with a system without adding a capture agent. The results are illustrated in Figure 15. The degradation rate of PNP is 92% when there is no trapping agent in the system. When the capture agent EDTA-2Na was added to the system, the degradation rate of PNP was unchanged, indicating that there is no significant role of h^+^ in the photocatalytic system. The degradation rate of PNP decreased significantly with the addition of IPA and BQ, and the inhibition effect of IPA was greater. Therefore, the capture experiment showed that both •OH and •O_2_^−^ in the FeOOH/g-C_3_N_4_ photo-Fenton-like system played roles in PNP degradation, with •OH having a more significant impact.

### 3.6. Comparison with Other Results about Modification of g-C_3_N_4_

The comparison of photocatalytic efficiency of other modified g-C_3_N_4_ is shown in Table 5. It can be seen that g-C_3_N_4_ is commonly modified by non-metals, metals and their oxides. The catalyst used in this experiment has a great advantage in time and degradation rate. 

## 4. Conclusions

In summary, FeOOH/g-C_3_N_4_ was successfully prepared by thermal polycondensation and coprecipitation, serving as a highly effective heterogeneous Fenton photocatalyst for the removal of PNP from water. The optimum preparation conditions were obtained as follows: g-C_3_N_4_ calcined at 600 °C for 4 h, with a mass fraction of FeOOH of 0.3. Single-factor experiments were conducted to establish the optimal process conditions for PNP degradation: H_2_O_2_ dosage of 4 mmol/L, catalyst dosage of 1 g/L, PNP initial concentration of 10 mg/L and pH value of 5. Under these conditions, the degradation efficiency of PNP by FeOOH/g-C_3_N_4_ reaches 92% within 120 min. Characterization techniques including SEM, XRD and UV-vis proved that FeOOH was successfully loaded on the surface of g-C_3_N_4_ without affecting the crystal structure of g-C_3_N_4_. After five cycles of photocatalytic degradation of PNP by FeOOH/g-C_3_N_4_, the degradation efficiency remained nearly 80%, which proves that the FeOOH/g-C_3_N_4_ photocatalyst prepared in this paper has good recycling and stability. Moreover, capture experiments revealed that both •OH and •O_2_^−^ play roles in the photocatalytic degradation of PNP by FeOOH/g-C_3_N_4_, while •OH plays a more significant role.

## Figures and Tables

**Figure 1 molecules-29-03202-f001:**
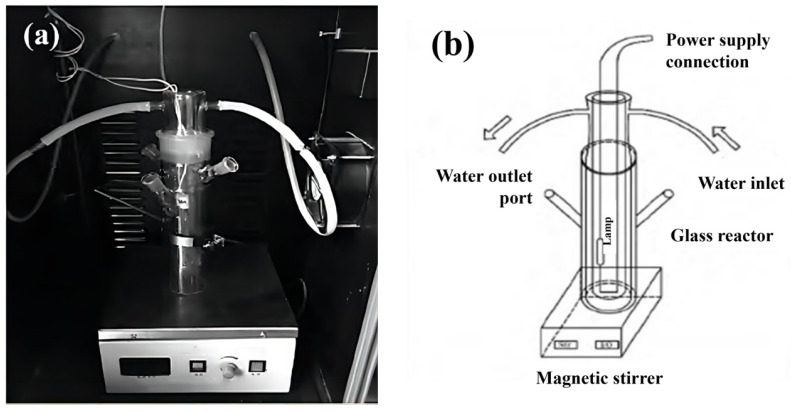
Equipment: (**a**) picture of real products and (**b**) schematic diagram.

**Figure 2 molecules-29-03202-f002:**
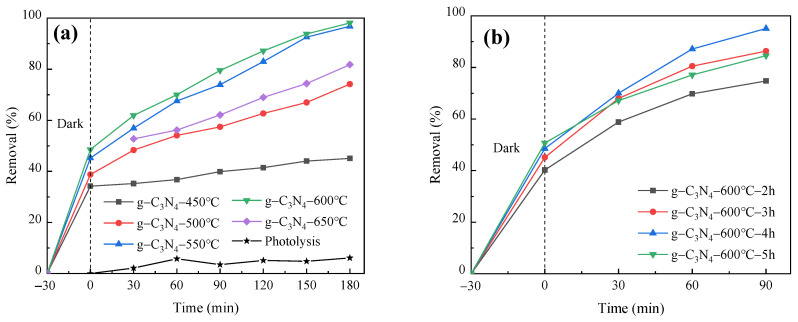
Effect of (**a**) calcination temperature and (**b**) calcination time on photocatalytic degradation of RhB by g-C_3_N_4_.

**Figure 3 molecules-29-03202-f003:**
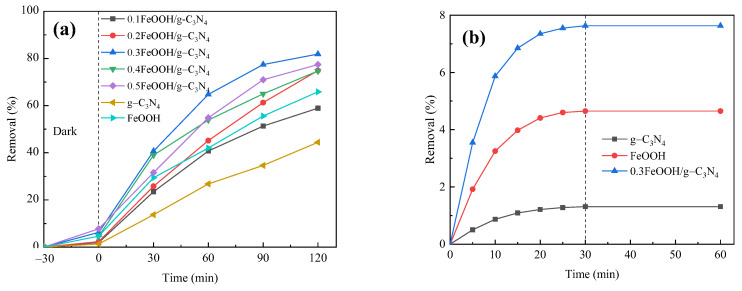
Effect of FeOOH mass fraction on photocatalytic degradation of PNP by FeOOH/g-C_3_N_4_: (**a**) total degradation rate and (**b**) dark adsorption curve.

**Figure 4 molecules-29-03202-f004:**
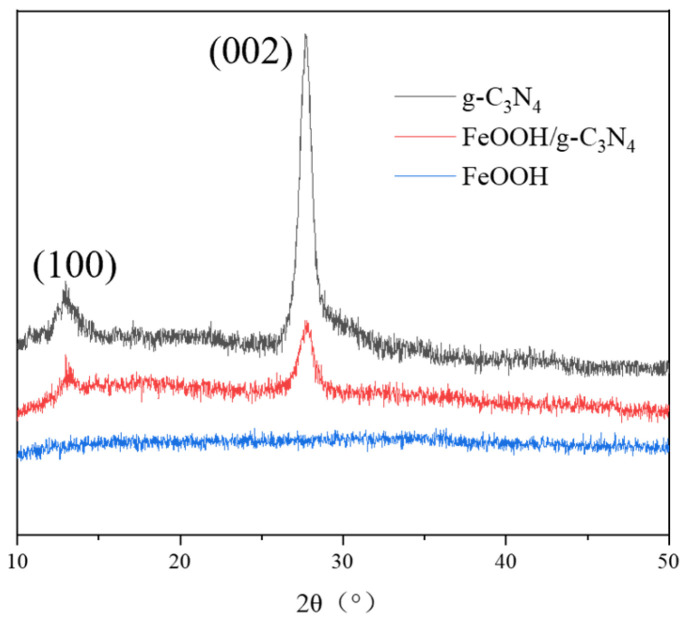
XRD patterns of g-C_3_N_4_, FeOOH/g-C_3_N_4_ and FeOOH.

**Figure 5 molecules-29-03202-f005:**
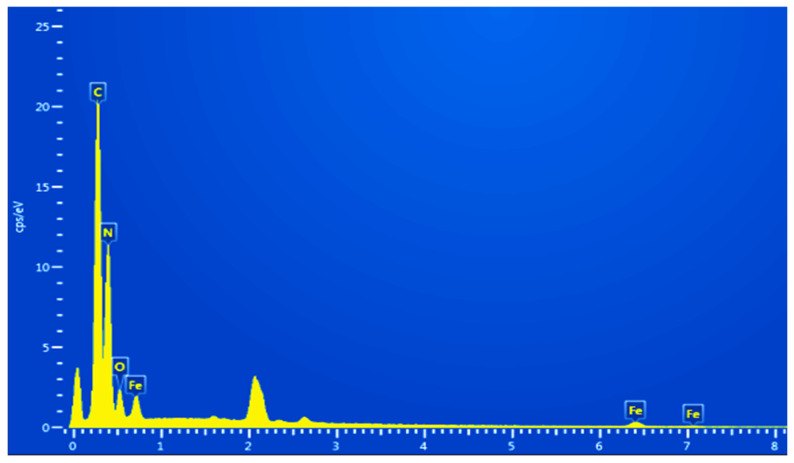
EDS diagram of FeOOH/g-C_3_N_4_.

**Figure 6 molecules-29-03202-f006:**
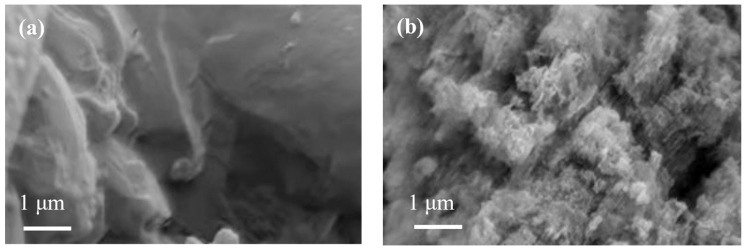
SEM of (**a**) g-C_3_N_4_ and (**b**) FeOOH/g-C_3_N_4_.

**Figure 7 molecules-29-03202-f007:**
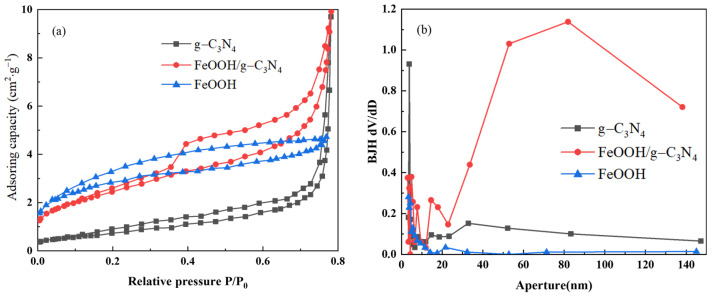
N_2_ adsorption–desorption curves (**a**) and pore size distribution curves (**b**) for g-C_3_N_4_ and FeOOH/g-C_3_N_4_ catalysts.

**Figure 8 molecules-29-03202-f008:**
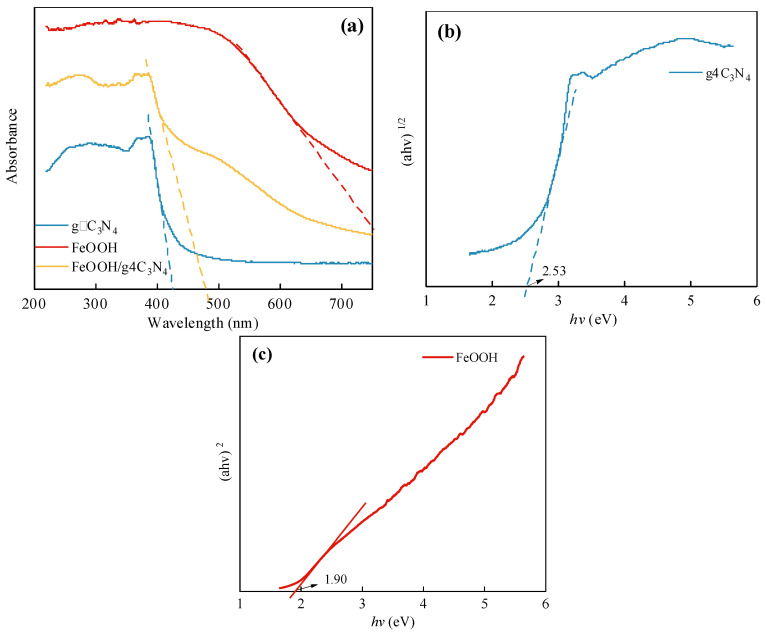
(**a**) UV-Vis DRS of g-C_3_N_4_, FeOOH and FeOOH/g-C_3_N_4_; the *Tauc plots* of (**b**) g-C_3_N_4_ and (**c**) FeOOH.

**Figure 9 molecules-29-03202-f009:**
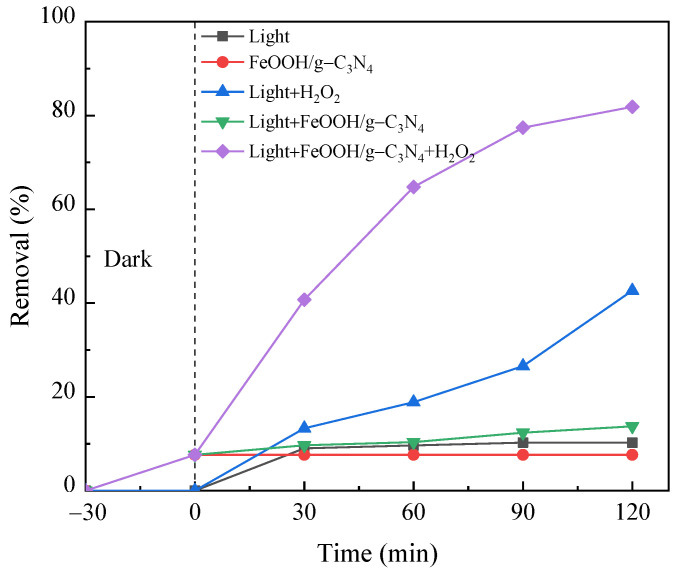
Effects of different degradation processes on PNP treatment.

**Figure 10 molecules-29-03202-f010:**
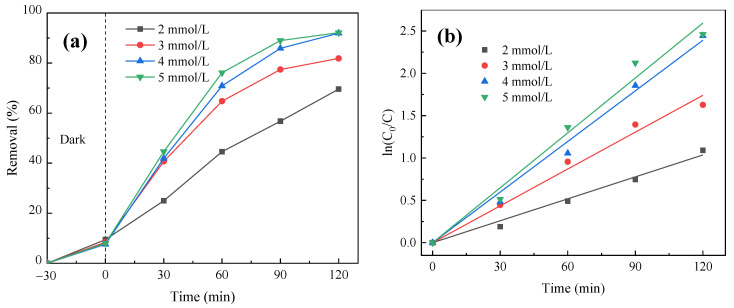
Effect of H_2_O_2_ concentration on PNP degradation efficiency: (**a**) photocatalytic degradation curve and (**b**) quasi-first-order kinetic fitting curve of photocatalytic reaction.

**Figure 11 molecules-29-03202-f011:**
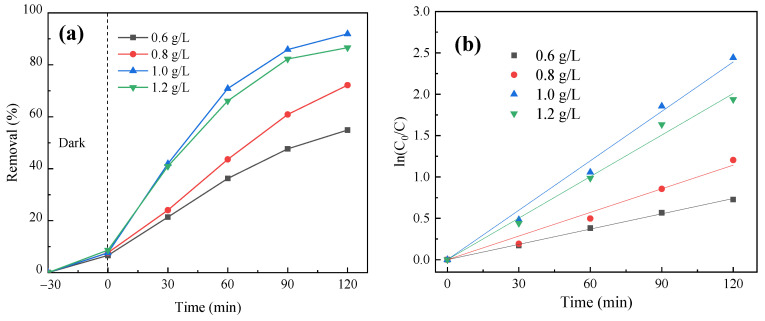
Effect of catalyst dosage on PNP degradation efficiency: (**a**) photocatalytic degradation curve and (**b**) quasi-first-order kinetic fitting curve of photocatalytic reaction.

**Figure 12 molecules-29-03202-f012:**
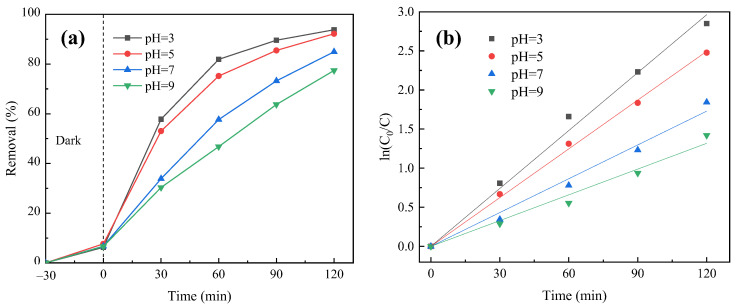
Effect of initial pH value of the reaction system on PNP degradation efficiency: (**a**) photocatalytic degradation curve and (**b**) quasi-first-order kinetic fitting curve of photocatalytic reaction.

**Figure 13 molecules-29-03202-f013:**
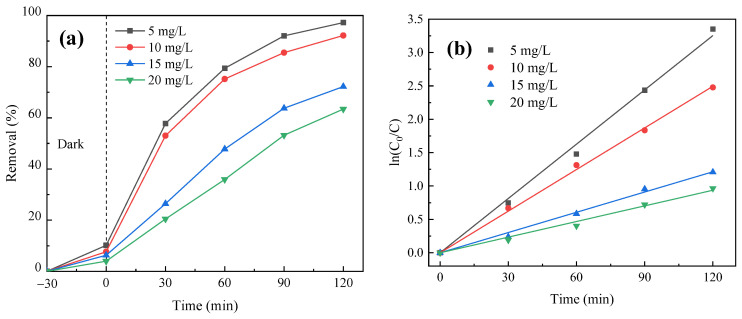
Effect of initial concentration of PNP on its degradation efficiency: (**a**) photocatalytic degradation curve and (**b**) quasi-first-order kinetic fitting curve of photocatalytic reaction.

**Figure 14 molecules-29-03202-f014:**
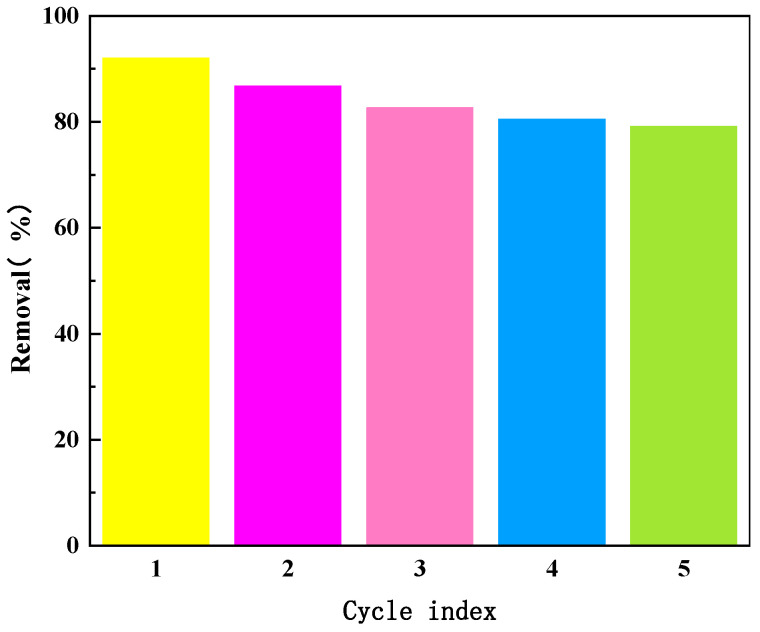
Recycling test of FeOOH/g-C_3_N_4_ photocatalyst for PNP degradation.

**Figure 15 molecules-29-03202-f015:**
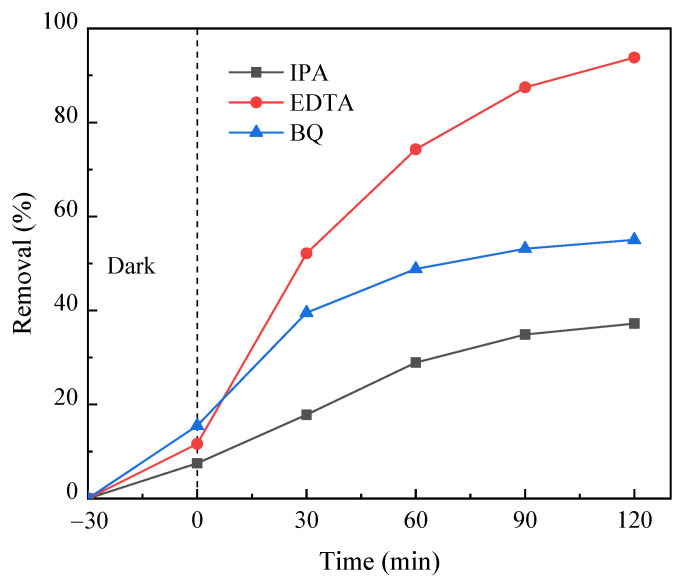
Active species capture experiment of PNP degradation by FeOOH/g-C_3_N_4_.

**Table 1 molecules-29-03202-t001:** Effect of H_2_O_2_ dosage on PNP degradation: quasi-first order kinetic analysis.

H_2_O_2_ Dosage (mmol L^−1^)	Kinetic Equation	Speed Constant *k* (min^−1^)	*R* ^2^
2.0	y = 0.00863x	0.00863	0.99412
3.0	y = 0.0145x	0.0145	0.99379
4.0	y = 0.01992x	0.01992	0.99543
5.0	y = 0.02161x	0.02161	0.99293

**Table 2 molecules-29-03202-t002:** Effect of catalyst dosage on PNP degradation: quasi-first order kinetic analysis.

Catalyst Dosage (mg L^−1^)	Kinetic Equation	Speed Constant *k* (min^−1^)	*R* ^2^
0.6	y = 0.00616x	0.00616	0.99924
0.8	y = 0.00952x	0.00952	0.99085
1.0	y = 0.01992x	0.01992	0.99543
1.2	y = 0.01674x	0.01674	0.99574

**Table 3 molecules-29-03202-t003:** Effect of initial pH on PNP degradation: quasi-first order kinetic analysis.

Initial pH Value	Kinetic Equation	Kinetic Equation *k* (min^−1^)	*R* ^2^
3	y = 0.02469x	0.02469	0.99628
5	y = 0.02079x	0.02079	0.99916
7	y = 0.01441x	0.01441	0.99292
9	y = 0.01098x	0.01098	0.9901

**Table 4 molecules-29-03202-t004:** Effects of PNP initial concentration on its degradation: quasi-first order kinetic analysis.

Initial Conc. of PNP (mg L^−1^)	Kinetic Equation	Kinetic Equation *k* (min^−1^)	*R* ^2^
5	y = 0.02712x	0.02712	0.99774
10	y = 0.02079x	0.02079	0.99916
15	y = 0.01011x	0.01011	0.99701
20	y = 0.00781x	0.00781	0.99443

**Table 5 molecules-29-03202-t005:** Comparison of degradation results of organic pollutants by different catalysts.

Catalysts	Target Pollutant	Light Sources	Degradation Time (min)	Degradation Rate (%)	Refs
Al_2_O_3_/g-C_3_N_4_	RhB	300 W Xe lamp	120	83	[38]
ZnO/g-C_3_N_4_	MO	150 W Xe lamp	300	65	[39]
O-g-C_3_N_4_	Cr(VI)	300 W Xe lamp	60	80	[40]
α-Fe_2_O_3_/g-C_3_N_4_	Cr(VI)	300 W Xe lamp	150	98	[41]
TiO_2_/g-C_3_N_4_	RhB	300 W Xe lamp	180	80	[42]
FeOOH/g-C_3_N_4_	brilliant blue	500 W Xe lamp	30	99	Present work

## Data Availability

The data presented in this study are available in article.

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
