# Peer review of "High-Efficiency Photo-Fenton-like Catalyst of FeOOH/g-C3N4 for the Degradation of PNP: Characterization, Catalytic Performance and Mechanism Exploration"

_molecules, 2024, doi:10.3390/molecules29133202_

Round 1
Reviewer 1 Report
Comments and Suggestions for Authors
The paper presents the synthesis and characterization of a FeOOH/g-C3N4 composite photocatalyst aimed at degrading p-nitrophenol (PNP) in water. The authors utilized thermal polycondensation and co-precipitation methods to prepare the catalyst, followed by extensive characterization techniques including XRD, SEM, and UV-Vis DRS. The study explores optimal preparation conditions and evaluates catalytic performance under various experimental parameters. The findings indicate that FeOOH/g-C3N4 demonstrates high degradation efficiency for PNP under visible light, with both •OH and •O2- playing significant roles in the photocatalytic process. However, minor revisions are required before acceptance.
1. The authors suggest in Figure 2a that the C/N ratio of g-C3N4 increases with rising temperature, leading to a decreased band gap. It is recommended to provide a UV-Vis spectrum showing the relationship between the C/N ratio and band gap to confirm if the reduced band gap is responsible for the increased degradation rate.
2. Figure 2b indicates that the prolonged calcination time enhances the specific surface area and pore volume of g-C3N4. This claim should be substantiated with BET data.
3. In Figure 6, the SEM images are mislabeled, with image (b) appearing first and the other image lacking a scale bar.
4. the description of the absorption edge in Figure 8 is incorrect and needs to be revised.
Author Response
To the referee’s comments, we make the following responses and changes in the manuscript:
- The authors suggest in Figure 2a that the C/N ratio of g-C3N4 increases with rising temperature, leading to a decreased band gap. It is recommended to provide a UV-Vis spectrum showing the relationship between the C/N ratio and band gap to confirm if the reduced band gap is responsible for the increased degradation rate.
Answer: Thanks for the reviewer’s suggestion. Due to the limited time for g-C3N4 preparation and characterization, we added corresponding references to prove this conclusion in the revised manuscript.
Manuscript: This result is because the C/N ratio of g-C3N4 increases with the increase in temperature, and the band gap decreases according to other references [33-34].
- Figure 2b indicates that the prolonged calcination time enhances the specific surface area and pore volume of g-C3N4. This claim should be substantiated with BET data.
Answer: Thanks for the reviewer’s suggestion. Due to the limited time for g-C3N4 preparation and characterization, we added corresponding references to prove this conclusion in the revised manuscript.
Manuscript: This phenomenon can be attributed to the increase in the specific surface area and pore volume of g-C3N4 with prolonged calcination time [35].
- In Figure 6, the SEM images are mislabeled, with image (b) appearing first and the other image lacking a scale bar.
Answer: The SEM images have been revised in the revised manuscript.
Manuscript:
Fig. 6. SEM of (a) g-C3N4 (b) FeOOH/g-C3N4.
- The description of the absorption edge in Figure 8 is incorrect and needs to be revised.
Answer: The description of the absorption edge in Figure 8 have been revised in the revised manuscript.
Manuscript: The absorption edge of pure g-C3N4 is about 430 nm, while that of FeOOH/g-C3N4 is extended to about 480 nm, which means it red-shifted and has a wide range, thus the absorption intensity is strengthened.

Reviewer 2 Report
Comments and Suggestions for Authors
Figure 1 should containt letters in figures and „Drawing“ in title should be deleted.
3.Results. and discussion L126 a “.” should be deleted
Figure 6. SEM of (a) g-C3N4 (b) FeOOH/g-C3N4. – figures should contain letters
Figure 9. in the figure latin letters should be used.
In general English should be checked. The manuscript contains many technical errors.
Author Response
To the second referee’s comments, we make the following responses and changes in the manuscript:
- Figure 1 should containt letters in figures and „Drawing“ in title should be deleted.
Answer: Thanks for the reviewer’s suggestion. The letters has been added in Figure 1 and the title has been revised in the revised manuscript.
Manuscript:
Fig. 1. Equipment: (a) picture of real products and (b) schematic diagram.
- 3.Results. and discussion L126 a “.” should be deleted.
Answer: The “.” have been deleted in the revised manuscript.
Manuscript: 3. Results and discussion
- Figure 6. SEM of (a) g-C3N4 (b) FeOOH/g-C3N4. – figures should contain letters
Answer: The letters have been added in Figure 6 in the revised manuscript.
Manuscript:
Fig. 6. SEM of (a) g-C3N4 (b) FeOOH/g-C3N4.
- Figure 9. in the figure latin letters should be used.
Answer: The latin letters have been used in the revised manuscript.
Manuscript:
Fig. 9. Effects of different degradation processes on PNP treatment.
- In general English should be checked. The manuscript contains many technical errors.
Answer: Thanks for the reviewer’s suggestion. Sorry for the grammar and spelling mistakes, we had corrected them using a language editing service. And we have made a thorough check in the format and grammar for the manuscript, which should meet the publication standard of this journal.

Reviewer 3 Report
Comments and Suggestions for Authors
I accept the manuscript to be published after major revision. The research work is interesting and carried out using developed techniques. However, some points must taken inconsideration before publication
1-The role of FeOOH in enhancing the photocatalytic degradation must explained in the introduction section accompanied by the role of FeOOH as photocatalyst in the recent previous research
2-Photolysis of rhodamine B dye in absence of the catalyst must carried out
3-The absorption spectrum for the photocatalytic degradation of rhodamine B and p-nitrophenol is required
4-The number of XRD card and the crystalline planes must inserted in XRD diagram
5-The adsorption isotherm of N2 must be discussed in term of type and pore structure
6-Tauc plot is required to determine band gap energy and type of transition
7-The valence and conduction band potential of the two semiconductors cand be termined from Kulblka-Munk equation
8-The scheme for charge transportation between the two semiconductor must illustrated with identification of the mechanism of the heterojunction construction whether type I or type II or Z-scheme
Comments on the Quality of English LanguageThe English language is good
Author Response
To the referee’s comments, we make the following responses and changes in the manuscript:
I accept the manuscript to be published after major revision. The research work is interesting and carried out using developed techniques. However, some points must taken inconsideration before publication.
- The role of FeOOH in enhancing the photocatalytic degradation must explained in the introduction section accompanied by the role of FeOOH as photocatalyst in the recent previous research.
Answer: Thanks for the reviewer’s suggestion. The role of FeOOH as photocatalyst in the recent previous research have been added in the revised manuscript.
Manuscript: FeOOH, with the advantages of narrow bandgap, broad visible absorption region, has been used for coupling with other semiconductors for applications such as direct photocatalytic degradation of organic matter. In addition, the band gap of amorphous FeOOH is much smaller than that of the corresponding crystalline materials, which results in a broader visible absorption region and excellent photocatalytic performance. However, there are few reports on amorphous FeOOH as a photocatalyst modifier.
- Photolysis of rhodamine B dye in absence of the catalyst must carried out.
Answer: The photolysis of rhodamine B dye in absence of the catalyst have been added in the revised manuscript.
Manuscript:
Fig. 2. Effect of (a) calcination temperature and (b) calcination time on photocatalytic degradation of RhB by g-C3N4.
- The absorption spectrum for the photocatalytic degradation of rhodamine B and p-nitrophenol is required.
Answer: We are very sorry, but due to time constraints and damage to the laboratory equipment, it was not possible to add this section. Our experiments were based on absorbance measurements and the data are reliable.
- The number of XRD card and the crystalline planes must inserted in XRD diagram.
Answer: The crystalline planes has been added in the revised manuscript. Since FeOOH is amorphous in this manuscript, there is no standard card.
Manuscript: XRD was used to characterize the crystal structure of the catalysis. As shown in Fig. 4, there are two distinct characteristic peaks located at 12.8° and 27.2°, which belong to the (100) and (002) surfaces of g-C3N4 respectively, indicating that g-C3N4 has a high purity phase (JCPDS No. 87-1526).
Fig. 4. XRD patterns of g-C3N4, FeOOH/g-C3N4 and FeOOH.
- The adsorption isotherm of N2 must be discussed in term of type and pore structure.
Answer: We have revised the discussion of adsorption isotherm of N2 in the revised manuscript.
Manuscript: As can be seen from Fig. 7a, N2 adsorption-desorption isotherms of g-C3N4 and FeOOH/g-C3N4 samples are consistent with the hysteresis characteristic curves of mesoporous materials with a type IV isotherm and a type-H3 adsorption hysteresis loop in the high P/P0 range.
- Tauc plot is required to determine band gap energy and type of transition.
Answer: We have added the Tauc plot in the revised manuscript.
Manuscript: For semiconductor materials, the band gap (Eg) is calculated as αhν = A(hν - Eg)n/2, where α is the absorption coefficient, h is Planck's constant, ν is the optical frequency, and A is a constant. n value is determined by the type of semiconductor leap. Since g-C3N4 is an indirect semiconductor, n is taken as 4, while FeOOH is a direct semiconductor, so n is 1. Fig. 8b-c displays band gaps of 2.53 eV and 1.90 eV for g-C3N4 and FeOOH, respectively.
Fig. 8. (a) UV-Vis DRS of g-C3N4, FeOOH and FeOOH/g-C3N4; The Tauc plots of (b) g-C3N4 and (c) FeOOH.
- The valence and conduction band potential of the two semiconductors cand be termined from Kulblka-Munk equation.
Answer: Thanks for the reviewer’s suggestion. The valence and conduction band positions are not directly obtained from the Kulblka-Munk equation. Since we did not test its VB-XPS, we cannot get the valence band and conduction band positions exactly.
- The scheme for charge transportation between the two semiconductor must illustrated with identification of the mechanism of the heterojunction construction whether type I or type II or Z-scheme.
Answer: Since we did not test its VB-XPS, we cannot get the valence band and conduction band positions exactly. The charge transportation between the two semiconductor follows a Z-scheme according to references.
Manuscript: The charge transportation between the FeOOH/g-C3N4 heterojunction follows a Z-scheme according to references [36-37]. Hence, electron-hole (e- -h+) pairs were generated on the surface of FeOOH and g-C3N4 photocatalysts under visible light irradiation. Briefly, the electrons are excited into the CB while the holes remain in the VB. Due to the formation of heterojunction, the FeOOH holes migrate downward. Conversely, the electrons of g-C3N4 will migrate to the more positively charged CB of FeOOH. As a result, the electrons gather on the CB of FeOOH and the holes gather on the VB of g-C3N4 for the purpose of photogenerated charge separation.
